# Development of an Information and Communication Technology (ICT) Tool for Monitoring of Antimicrobial Use, Animal Disease and Treatment Outcome in Low-Income Countries

**DOI:** 10.3390/antibiotics14030285

**Published:** 2025-03-10

**Authors:** Susanna Sternberg-Lewerin, Joshua Orungo Onono, Sofia Boqvist, Lawrence Mugisha, Wangoru Kihara, Linnea Lindfors, Kajsa Strandell, Florence Mutua

**Affiliations:** 1Department of Animal Biosciences, Swedish University of Agricultural Sciences, SE-75007 Uppsala, Sweden; sofia.boqvist@slu.se; 2Department of Public Health, Pharmacology & Toxicology, University of Nairobi, P.O. Box 30197, Nairobi 00100, Kenya; joshua.orungo@uonbi.ac.ke; 3Department of Wildlife and Aquatic Animal Resources, Makerere University, P.O. Box 7062, Kampala 999123, Uganda; lawrence.mugisha@mak.ac.ug; 4Badili Innovations Ltd., Nairobi 00100, Kenya; wangoru.kihara@badili.co.ke; 5Veterinärerna i Svenljunga, SE-51293 Svenljunga, Sweden; linnea.lindfors@hotmail.com; 6Falkenbergs Veterinärservice, SE-31150 Falkenberg, Sweden; kajsa.strandell@gmail.com; 7International Livestock Research Institute, P.O. Box 30709, Nairobi 00100, Kenya; f.mutua@cgiar.org

**Keywords:** AMU, AMR, monitoring, e-surveillance, veterinary pharmaceuticals, LMICs

## Abstract

Background/Objectives: Surveillance of antimicrobial resistance (AMR) and antimicrobial use (AMU) is needed to understand risks and implement policies. Collecting AMU data in the context of disease prevalence and therapeutic outcomes has been suggested for improving AMU. We describe the process of developing an information and communication technology (ICT) system to monitor AMU, diseases and treatment outcomes in poultry in East Africa. Methods: A prototype system to register drug sales in veterinary pharmacies, a mobile application for farmers to report their AMU, and a database for monitoring was developed. Contact information for participating veterinarians was included as well as information about poultry diseases, husbandry, AMR and prudent use of antibiotics. The system was pilot-tested for a 6-month period in Kenya. Results: A total of 15,725 records were submitted by the 14 participating pharmacies and 91 records were entered by the 15 participating farmers. Overall, the participants were positive about the system and were able to use it. The information available was appreciated by the farmers. The pharmacy representatives appreciated getting an overview of their sales and stated that it had given them new insights. Conclusions: Despite some challenges, the concept of the developed ICT system could be useful for future monitoring of animal health and the use of pharmaceuticals in animals, and connecting farmers with veterinarians to improve animal health management. Our results underline the importance of close collaboration with stakeholders so that developed tools can be transferred to national ownership after the finalization of externally funded projects.

## 1. Introduction

Antimicrobial resistance (AMR) is a global health threat. In addition, resistant bacteria in livestock production presents a threat to global food production. It is well known that antimicrobial use (AMU) exerts a selective pressure for the development and emergence of resistant microbes, and the non-medical use in animal production has therefore been banned in many regions, such as the European Union [1]. Nevertheless, in many low- and middle-income countries (LMICs), antimicrobial drugs are given to healthy animals for disease prevention and growth promotion, as well as for treatments without consulting an animal health professional [2]. Surveillance of AMR and AMU is needed to understand specific risks and implement mitigation policies, and the World Organization for Animal Health (WOAH) has been collecting data on the amounts and reasons for antimicrobial use in animals since 2015, mainly based on sales data [3]. However, there is no global harmonization of the collection of AMU data [4] and specific AMU data in different livestock species are lacking in most LMICs, which hampers policymaking to mitigate AMR risks [5]. Animal disease surveillance complements AMR/AMU surveillance and provides additional insights, but there is also a need to understand the reasons for specific treatments and treatment failures, to design sustainable disease management policies. Collecting AMU data in the context of disease prevalence and therapeutic outcomes has been suggested as useful for improving AMU [6]. In summary, there is a need for integrated surveillance of AMU and AMR, and improved animal disease management in many LMICs. Treatment failures could be used as a proxy for AMR, where laboratory-based surveillance is not yet available.

Poultry production is the fastest-growing sector in animal production globally and in East Africa [7] and a parallel increase in AMU in this sector is predicted [4]. In many LMICs, there is no requirement for a veterinary prescription for AMU and it has been reported that poultry farmers in LMICs often turn to drug sellers or fellow farmers instead of consulting a veterinarian for advice on AMU [8].

In Kenya, veterinary pharmacies are licensed by the Veterinary Medicines Directorate (VMD) and must be managed by qualified veterinary professionals [9]. Similarly in Uganda, veterinary drug outlets are registered by the National Drug Authority (NDA) [10]. A previous study on antibiotic retailers in Nairobi revealed that not all staff working as veterinary pharmacists had the required education, only 57% had clinical training and only 41% had a formal education in AMU [11]. Current legislation has made it mandatory to have a veterinary prescription for the purchase of antibacterial drugs (hereafter referred to as antibiotics). It has, however, been shown that Kenyan poultry farmers reportedly purchase antibiotics without veterinary prescriptions [12]. The VMD has recently launched a national campaign to enforce compliance with the current legislation and other requirements for veterinary pharmacies. Pharmacies are obliged to record their sales of antimicrobials, and this record-keeping usually relies on hand-written notes in a ledger. Pharmaceuticals are registered, as other goods, in the government system for imports and exports but there is currently no system for collecting AMU data at various distribution and retail nodes along the marketing chains and at the farm level in Kenya.

For animal disease reporting, the Kenya Animal Biosurveillance System has been implemented, allowing animal health professionals to fulfill their obligation to report notifiable animal diseases via their smartphones and the Kenya Directorate of Veterinary Services to continuously collect and analyze data on animal disease occurrences [13]. Mobile applications offer opportunities to collect data from multiple stakeholders in real-time, provided there is sufficient access to mobile phones and internet access in the target population. Kenya is one of the leading countries in Africa in smartphone usage [14], and mobile applications are attractive tools for data collection that can be integrated into existing systems [15]. This paper focuses on the process of developing an information and communication technology (ICT) system to monitor pharmaceutical use, diseases and treatment outcomes in animals in LMICs. In addition to the general description, the information included for the initial target group, poultry production, is provided. The work was an integral part of the MAD-tech-AMR (management of animal diseases and antimicrobial use by information and communication technology to control AMR in East Africa) project funded by the European Joint Programming Initiative on Antimicrobial Resistance (https://www.jpiamr.eu/projects/mad-tech-amr/, URL accessed on 9 March 2025), initially conducted in Kenya and Uganda. The objective was to provide a proof-of-concept for an ICT system to support improved animal health management, responsible AMU and reporting of AMU. The process aligned with the recently published FAO and WOAH guidelines on monitoring antimicrobial use at the farm level [16], starting on a small scale with one animal species and with continuous collaboration with relevant stakeholders. The current policy landscape in Kenya provided an opportunity to test this.

## 2. Results

The stakeholder consultations revealed some specific wishes and needs, but did not prompt any substantial changes to the content of the system. On the other hand, input concerning how and when the intended users would be expected to operate the system was discussed within the project team and taken into account as far as possible, both in the design of the system and the setup of the pilot study.

Out of the 100 participating farmers in the baseline study, 15 took part in testing the system in the pilot study. In addition, all 14 veterinary pharmacies from the baseline study participated in the pilot testing of the system.

The focus group discussions (FGDs) that were conducted to capture the perceptions of the pilot study participants included most, but not all, participants. The FGDs three months into the pilot study included 11 of the participating farmers in two FGDs (one in Machakos and one in Kajiado) and 12 representatives from the veterinary pharmacies, also in two separate FGDs (one in each county). The FGDs conducted after the end of the pilot included 10 of the representatives from the veterinary pharmacies and 9 of the poultry farmers, in separate FGDs for each participant group and each county (i.e., a total of four FGDs on each occasion).

After three months, eight of the eleven farmers in the FGDs had been using the ADIS system on their own, while three had not used the system after the introduction, due to technical issues (mainly poor internet stability). All those who had been browsing the system were generally positive about it and found it useful. One farmer praised the system for gathering easily accessible information onto a single platform, reducing the necessity to contact veterinarians. Another described it as eye-opening and that the ADIS system had been providing new insights into, for example, vaccination against certain diseases. The farmers found the disease illustrations and contact information for veterinary pharmacies on the website useful. The frequency of browsing the information and reporting of diseases and treatments depended on the health status of the animals. During the study period, farmers who experienced good health in their flock used the platform infrequently and reported no treatments, while some mentioned unreported treatments due to varying usage of the system. Challenges in reporting treatments were mainly related to difficulties in understanding the medicine list in the dropdown menu. The farmers suggested including information about more diseases, indigenous medicines and a list of treatments for different diseases. An interactive part, where farmers could exchange experiences and discuss treatments, was also suggested. Technical issues were the most prominent problems mentioned, such as incompatibility of phones using different operating systems (i.e., iPhone), issues with network and screen display, page loading and login difficulties. However, many of the farmers stated that they received assistance from the project’s IT team that promptly addressed any reported technical issues. In the final FGDs, the farmers stated that they had been able to report diseases as well as treatments in the system. No additional perceptions of the system emerged in these discussions; appreciation of the information that could be retrieved was repeated. A total of 91 records were entered by the 15 participating farmers.

On both occasions, the FGDs with representatives from the veterinary pharmacies yielded similar results. Overall, they expressed satisfaction with the system, stating that the tool worked well. Initially, the impression that sales needed to be registered instantly caused disruptions during peak hours. Subsequently, alternative strategies for registering sales were adopted. Several participants encountered technical challenges during the initial days of use, but these were quickly resolved by the IT support team (as the project could capture the issues raised and respond to them). Another early issue was the absence of some drugs in the system, this was also solved. At both three months and at the end of the pilot, there were comments about the system having provided them with new insights about their sales of antibiotics, and some stated that the system facilitated follow-up of the treatment outcome (recovery or not) for the prescribed drugs with the customers. A few reported having accessed the disease information module. Some said that the system helped them to obtain the correct disease history from the farmers which at times led to a reduction in antibiotic use and consequent reduction in sales. At the end of the pilot study, a total of 15,725 records had been submitted by the 14 participating pharmacies. Reportedly, the percentage of sales captured ranged between 75 and 100%. Most of the participating pharmacy representatives stated that they would be interested in using the system in the future, for record-keeping as well as to list their customers, and to track drug sales. They believed that the infographic would be helpful to their customers and suggested presenting it on a poster for the farmers to see when they visit the pharmacies.

In the final stakeholder meeting held in August 2024, preliminary project results were discussed with stakeholders from the local and national veterinary authorities. After demonstrating the ICT system, most reactions were positive. The system was perceived as user-friendly, something that was emphasized as a necessary aspect if it was to be implemented. It was stated that collection of AMU data is needed, and implementation of the ADIS system itself, or another system based on the same concept, was discussed. The inclusion of disease information was questioned by some participants, as it was seen as encouraging farmers to self-medicate, while others said that such information can be obtained online anyway and therefore it was better to include it in a format that encourages farmers to seek veterinary advice.

The current government system only records data on antimicrobials that have been imported or exported from the country, with no recording of how antimicrobials that are imported reach the numerous retail veterinary pharmacies through the wholesalers. Therefore, a system that records data on wholesalers and/or pharmacies was seen as useful to help the VMD trace the various molecules to the end users.

As Kenyan legislation and policy development have created a context where the concept of ADIS and the aim of the MAD-tech-AMR project are useful, initiatives building on these were discussed. One action included designing a training program for staff in veterinary pharmacies, and another creating an information package for farmers that could be distributed in veterinary pharmacies. These activities have been carried out during October–November 2024, in collaboration with the local and national authorities. Ultimately, these actions aim to improve responsible AMU and manage AMR in Kenyan livestock production.

## 3. Discussion

The development of ICT tools entails many challenges, and these are exacerbated in LMICs where electricity supply and internet connectivity can be unstable. In addition, the types of mobile devices vary, and the situation is exacerbated by the fragmented Android ecosystem. This may, among other things, lead to login and page loading failures that cause frustrations among users. To incentivize people to spend time on reporting is also challenging. Nevertheless, the users of the ICT system were generally appreciative of it. If ADIS or a similar system were implemented on a national level, pharmacies could replace their current hand-written records with the digital system, facilitating both book-keeping, stock-taking and official controls. Even in the absence of a mandatory system, pharmacies might see these benefits as an incentive to move to digital record-keeping. The participating staff in veterinary pharmacies appeared to use the system more easily than the farmers who participated in the pilot testing. As most farmers did not experience many disease events during the study, they did not develop much experience in its use, while the pharmacy staff used the system continuously and were able to address any problems encountered with the support of the IT team. Some of the challenges experienced could be addressed if the system was further developed, for example providing the mobile app in both Android and iPhone versions. The issue with the medicines list that was difficult for the farmers to use arose from the display of the drug list, as the intention was to provide a dropdown list with product names and available packages that would be easily recognizable by the farmers, but the names of the drugs as registered by Kenya’s VMD and Uganda’s NDA proved to be inconsistent with the local names the farmers used. In the future, adapting the drug list display for the farmers would probably improve usability for this group.

During the limited project implementation period, it was not possible to assess any changes in behavior in regard to AMU and disease prevention. The true proportion of drug sales and on-farm treatments captured by the system could not be assessed, as the only data available were those provided by the pharmacies and farmers. Only 15% of the farmers from the baseline study participated in the pilot testing of the system, while all veterinary pharmacies from the baseline took part in the pilot. The numbers were seen as appropriate for the study, as regular follow-up visits to provide technical support were foreseen. It is questionable whether the participating farmers are representative of the general farmer population. However, the main objective was to test if the concept of ADIS could be used and to capture ideas on how to move forward to create and implement a monitoring system, not to study the behavior of the farmers. Metrics on user satisfaction were not available due to the qualitative nature of data collection in regard to this aspect. As the piloted system can be seen as a prototype rather than a ready-to-use tool, the qualitative approach was seen as more useful than a quantitative evaluation. However, if ADIS or a similar system is implemented on a larger scale, continuous evaluations would be needed as a basis for improving uptake, ensuring data quality and monitor impact. It would also need to be expanded to encompass all livestock species, to be useful as a national monitoring system. In addition, an offline mode might be needed to enable registration in areas with a poor internet connection.

We believe that the combination of monitoring pharmaceutical sales, animal treatments and treatment outcomes can be used as a stepping-stone towards more advanced AMR/AMU monitoring. Although data from pharmacies and farmers cannot be linked, a future system applied on a national level could use the geographic information related to the input data (pharmacy location and farmer location) to assess AMU on a regional level. In the long term, combining data on antimicrobial sales and drugs administered on the farm level can support national authorities by providing an overview of AMU, both in regards to the amounts used, the reason for use and the outcome of this use. Such data on a national and regional level will help design policies (legislation and control as well as information campaigns).

The in-built educational element about animal husbandry, animal diseases and AMR may serve as an incentive as well as knowledge dissemination. The appreciation of this element expressed by the participants seems to support this assumption. However, long-term compliance with reporting would need to be ensured for the system to serve its purpose, and this would likely require some policy intervention. Our hope was that the farmers would see the benefits of the information and veterinary contacts so that they would regularly use the system. Our recruited farmers were all literate and fluent in English. In the future, to enable widespread use of this type of system in LMICs, translation into local languages and an AI voice generator for illiterate users would likely be needed. In addition, regular input of data from system users could be a requirement for access, or put into legislation, to ensure continuous use of (and familiarity with) the system.

The current policy landscape in Kenya is a timely opportunity for this type of intervention. Nevertheless, the importance of engaging the right persons became obvious as interest from the authorities improved over time as the relevant staff became involved.

Initially, the intention was to include a telemedicine component, where farmers would consult veterinary professionals via video calls. This, however, proved too ambitious in the first phase and we decided to begin with providing contact details to veterinary professionals and encourage farmers to make use of their advice. However, data on the number of veterinary consultations via the system were not collected. There is currently no legal framework for telemedicine in the veterinary sector in either of the countries. Communication tools such as WhatsApp are frequently used in the region and could serve as a channel for remote consultations. The farmers seemed to appreciate the contact information for available animal health professionals.

Some farmers wanted treatment advice to be available in ADIS, to reduce the need for veterinary consultations, indicating that the fear of some veterinarians that farmers would use the system as a replacement for veterinary consultations was justified. As the purpose was the contrary, to increase veterinary consultations, the information in the system reflected the difficulties in diagnosing diseases and the importance of using pharmaceuticals properly, for cost-efficiency and safety. The emphasis was on good animal husbandry and disease prevention, and the risk of AMR and/or treatment failures if veterinary drugs were used irresponsibly. The information had a dual purpose: to provide farmers with useful advice that would improve their animal health management and to incentivize their use of the system. The farmers in the focus groups also wanted support for discussions between farmers, and this was also part of the original plan. So-called farmer field schools/stable schools can be a useful tool for capacity building [17,18]. However, their success requires a knowledgeable resource person and/or reliable information that can form the basis of new knowledge so that participants do not mislead each other and risk exacerbating problems rather than solving them. Hence, if the ICT system were to contain a forum for virtual discussions between farmers, this would need moderating by an animal health professional.

## 4. Materials and Methods

A prototype platform to register drug sales in veterinary pharmacies, and a mobile application for farmers to report about animal disease and veterinary drug use, linked to a database to monitor drug sales and provide information was developed. Pharmacies enter data based on drug and drug packages, whereas farmers enter data based on individual animal treatments. It was assumed that the farmers would only have access to a mobile phone, and as Android phones are less expensive, and therefore more common, this format was chosen. A web-based application would be easier for large amounts of data registration, therefore, this was the preferred option for veterinary pharmacies. The system was named ADIS (Animal Disease Information System), to emphasize the intention to improve animal health by including basic information about animal health management, responsible AMU, and AMR. Contact information for participating veterinary professionals was also included, to provide participating farmers with an opportunity for video consultation via WhatsApp or similar and paving the way for veterinary telemedicine to facilitate veterinary consultations. In addition, information about the most common poultry diseases in the region was included, as well as information about animal health management, AMR and prudent use of antibiotics. The prototype system is available at https://madtechamr.ilri.org/#/, (URL accessed on 9 March 2025) and a conceptual overview of the system is provided in Figure 1, and the user interfaces are shown in Figure 2.

A study was conducted in 2021, to collect baseline data about practices around sales and use of antibiotics in poultry farms in the two counties of Machakos and Kajiado, Kenya [12]. A parallel study was carried out in Uganda (Mugisha et al., manuscript in preparation). The results from these studies were used to identify relevant topics to be included in the information for ADIS users, to support improvement of AMU practices and animal health management. The results also provided useful information about the pharmaceutical value chain and its actors in Kenya, as well as an opportunity to recruit participants in the pilot testing of the ADIS system.

### 4.1. Technical Specifications

The ADIS system is based on the client-server architecture, which was selected due to the need to have a central database to store and manage the data from different client applications. ADIS consists of a mobile (Android) application for data collection from farmers, a web-based application for reporting by veterinary pharmacies, a backend application for processing and storing the collected data and an application programming interface (API) for data exchange between the frontend applications and the backend server (see Figure 3).

The mobile application was developed using Java SE18 (www.java.com, URL accessed on 9 March 2025) programming language and targeted Android devices running Android Jelly Bean with API level 17 and above. The web-based application was developed using React (react.dev) version 16.13.1. React is a JavaScript library for building user interfaces and targeting applications that are used from a computer browser. The backend server was developed using Python (www.python.org, URL accessed on 9 March 2025) version 3.12.4 and Django (www.djangoproject.com, URL accessed on 9 March 2025) version 3.1.4. MySQL (dev.mysql.com, URL accessed 9 on March 2025) version 8.0 LTS was used as the database server while the API was developed using the DjangoRest Framework (django-rest-framework.org, URL accessed on 9 March 2025) version 3.12.1. The user interfaces were finally rendered using HTML and CSS.

ADIS complies with local and international data protection regulations. Robust security measures are implemented to ensure the protection of data integrity and personally identifiable information (PII). The Linux-based server hosting the system is located within a secure, access-controlled room at the International Livestock Research Institute (ILRI), with access restricted exclusively to authorized system administrators. A virtual machine (VM) was configured on the Linux server to host the ADIS application, with access limited to whitelisted public IP addresses or devices connected to ILRI’s internal network. Both the VM and the backend database were secured using strong, complex passwords to mitigate unauthorized access. Participating farmers and veterinary pharmacies enter data based on informed consent and use their own personal passwords. There are different levels of access so that users can only access their own data. Overviews are only available to the administrator role and this role is limited to a few people.

### 4.2. Information Provided in ADIS

In ADIS, information is provided on-farm biosecurity, good animal husbandry practices, disease prevention, common diseases and AMR. The messages in the information are written as clearly and simply as possible, in brief points and organized so that specific information is easy to locate when seeking it. The information focuses on poultry, as commercial poultry farming was the initial target group. The animal husbandry section contains general disease-preventing measures such as hand washing and dedicated clothes as well as avoiding the introduction of sick birds and vaccination. It also includes general recommendations around temperature, feed and water, the safety of buildings, management of day-old chicks and animal welfare linked to production performance. The disease management information covers how to recognize signs of disease and what characterizes healthy birds, justifications for contacting a veterinarian/animal health service provider before drug treatments, explaining the importance of withdrawal periods, safe handling of drugs and disposal of drug waste. The diseases included are those commonly reported in East African poultry: Avian influenza, Fowl pox, Infectious bronchitis, Marek’s disease, Newcastle disease, Gumboro disease, Fowl cholera and Infectious coryza. The main clinical signs are listed, next to pictures sourced from the picture book of infectious poultry diseases [19], along with information about when antibiotics are not expected to work because the infectious agent is a virus. The messages align with the promotion of contacting a professional animal health worker; the diseases are not easy to distinguish from each other and a mistake may lead to the wrong treatment and failure to control a disease outbreak. The AMR information covers simple explanations about the problem, how it emerges and what are the consequences, along with an infographic (see Figure 4).

A list of registered drugs from the VMD in Kenya and the NDA in Uganda was loaded into the system to facilitate the recording of drug sales and drug use by farmers and veterinary pharmacies.

### 4.3. Stakeholder Consultations

Stakeholder consultations and workshops were conducted continuously as the system was being developed, to maximize user-friendliness and usefulness. The first stakeholder consultation workshops in Kenya were conducted online during the COVID-19 pandemic, in August and September 2021, and included representatives from the VMD, the County Directorates for Veterinary Services in Machakos and Kajiado, veterinarians with experience from veterinary pharmacies and poultry farmers. During these workshops, findings from the baseline work were presented and participants were offered an opportunity to suggest what they thought needed to be incorporated in the design of the ICT system. The initial framework was updated based on feedback from stakeholders and consultations within the project team. It was seen as crucial to include incentives to promote the use of the system. The animal health management information was intended for farmers but also accessible to the participating veterinary pharmacies so that they would know what the farmers could read, and also use the information for updating their own knowledge. In addition, the provision of contact information to facilitate farmers to call veterinary professionals in the participating pharmacies was intended as an incentive for both groups.

The second stakeholder consultation workshops were conducted in person, in June 2022 (for Kenya) and in October 2022 (for Uganda). The Kenya component consisted of two parts, one for study participants (farmers and representatives from veterinary pharmacies), and a separate one for other stakeholders including participants from the Directorate of Veterinary Services, County Veterinary and Livestock Departments (Machakos and Kajiado), VMD, pharmaceutical companies and feed industry. In Uganda, one workshop gathered all stakeholders, poultry farmers, government veterinary extension workers, veterinary drug shop owners and policy representatives from the Ministry of Agriculture, Animal Industry and Fisheries and the NDA. The meetings were an opportunity to demonstrate the tool and give stakeholders time to offer suggestions for improvement.

The system was finalized considering inputs from all the stakeholders, and recommendations from the project team. All ADIS users are informed about data protection, what data are collected and how they are handled and stored. Each user can only access their own data, linked to their login details and including data summaries, while administrators can extract data in spreadsheet format.

### 4.4. Pilot Testing of the System

During 2023, a 6-month pilot study was run in Kenya, with farmers and veterinary pharmacies that had participated in the baseline study. The demographics of this population are described in Mutua et al. [12]. No personal information about the participants in the pilot study was collected, to avoid deterring them from participation. They were invited to test ADIS, provided with IT support and reimbursed for the time they spent on reporting in the system. For farmers, having a smartphone with internet access was an inclusion criterion. Regular visits to all participants were made to monitor progress and respond to concerns raised or issues that the research team had observed in the database. Participants were encouraged to contact the project in case they encountered any challenges. A log was created to keep track of the issues reported and how they were managed. Halfway through the study, focus group discussions (FGDs) were conducted with the participants, separate groups with farmers and representatives from veterinary pharmacies [20,21] in each of the counties. At the end of the study period, data were extracted from the system and analyzed (Mutua et al., manuscript in preparation). A final data collection was undertaken, in the form of separate FGDs with the farmers and representatives from veterinary pharmacies in each county, about two months after the pilot phase ended. All FGDs were led by a moderator based on a checklist, with an assistant taking notes. In addition, the meetings were voice-recorded based on signed informed consent.

### 4.5. Follow-up Stakeholder Engagement

Further stakeholder workshops were conducted in Kenya, in April 2024, on separate days for study participants (and separately for farmers and representatives from veterinary pharmacies) and policymakers on a regional and national level. The meetings were intended to share findings from the pilot activity and to obtain feedback from the participants. Another workshop was performed in August 2024, with representatives from the veterinary departments at the county and national level to, among other things, discuss how the tool could support ongoing surveillance activities.

## 5. Conclusions

Despite some challenges, the concept of the developed ICT system could be useful for the future monitoring of animal health, use of antibiotics and other pharmaceuticals in animals, as well as connecting farmers with each other and with animal health professionals, to improve animal health management. Our results underline the importance of close collaboration with national and local stakeholders so that developed tools can be transferred to national ownership after the finalization of externally funded projects.

## Figures and Tables

**Figure 1 antibiotics-14-00285-f001:**
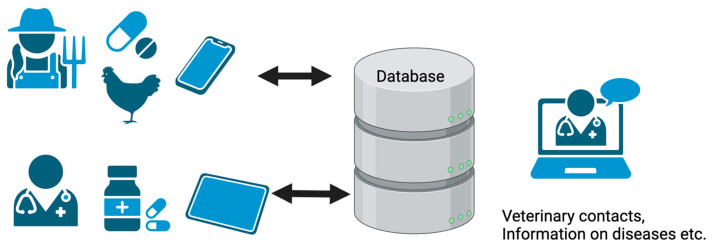
Conceptual overview of the ADIS system and its users. Farmers enter data on animal disease and medical treatment on their farm via a mobile app while veterinary pharmacies enter data on sales of pharmaceuticals via tablets. Each actor can retrieve a summary of their own data and access the information about animal diseases. Farmers can also obtain contact information for veterinary consultations. Overviews of all data entries can be accessed by the system administrator.

**Figure 2 antibiotics-14-00285-f002:**
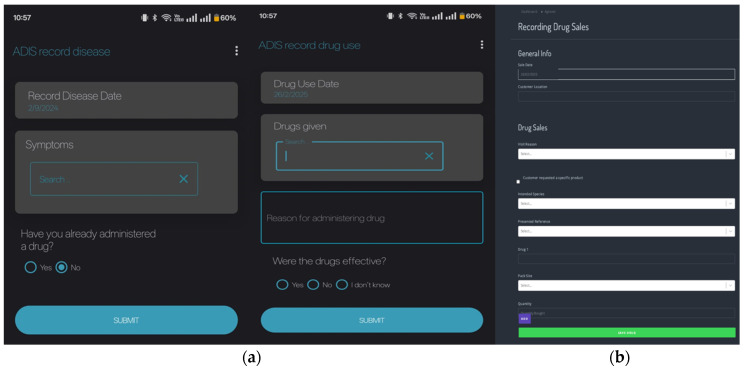
Screenshots of the ADIS system recording views for (**a**) farmers and (**b**) veterinary pharmacies.

**Figure 3 antibiotics-14-00285-f003:**
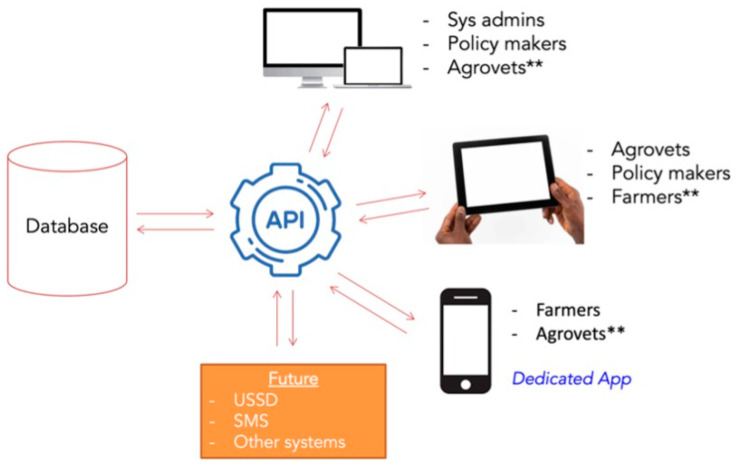
The ADIS system architecture. ** In the current system, farmers use the mobile phone App, Agrovets (veterinary pharmacy staff) use tablets to access the web application and administrators access data via computer.

**Figure 4 antibiotics-14-00285-f004:**
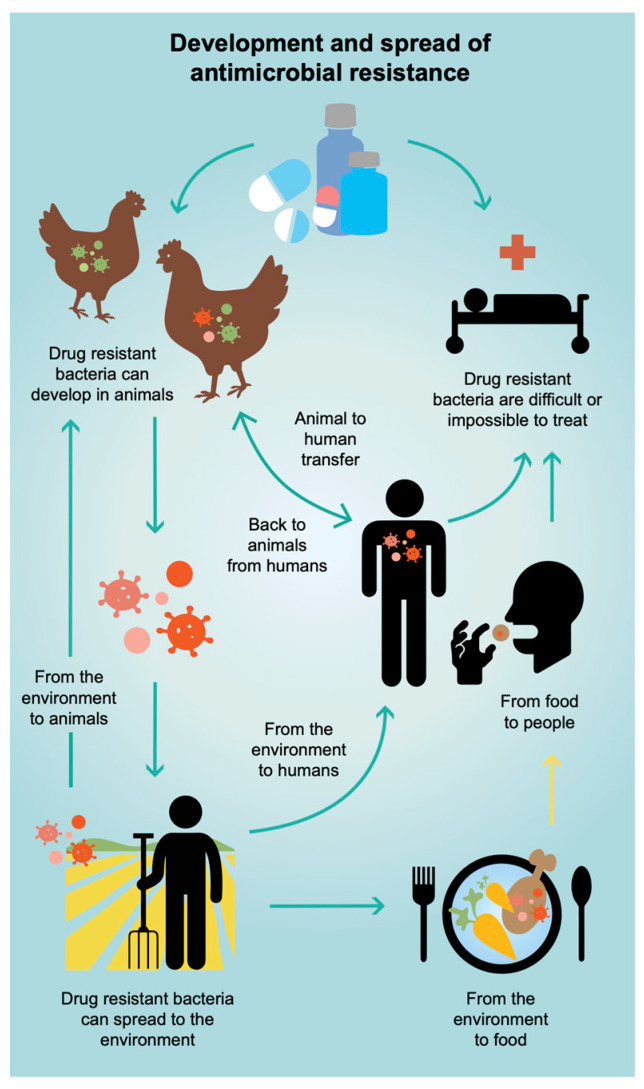
Infographic about AMR included in ADIS. Created by Cajsa Lithell, Redcap Design.

## Data Availability

The data generated during the current study are not publicly available due to a promise of anonymity to the study participants but will be published in summary format. Computer codes are available from the corresponding author on reasonable request.

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
