# Peer review of "Development of an Information and Communication Technology (ICT) Tool for Monitoring of Antimicrobial Use, Animal Disease and Treatment Outcome in Low-Income Countries"

_antibiotics, 2025, doi:10.3390/antibiotics14030285_

Round 1
Reviewer 1 Report
Comments and Suggestions for Authors
The manuscript presents a well-structured study on the development and testing of an ICT tool for monitoring antibiotic use, animal diseases, and treatment outcomes in low-income countries. The content is clear, the methodology is appropriately described, and the results provide valuable insights in a veterinary context.
I think that the manuscript is suitable for publication in its current form; however, some concerns remain regarding the article type. Manuscript is classified as a “Project Report”, while Antibiotics primarily accepts original research articles, review articles, short communications, and case reports. It would be advisable to confirm whether “Project Report” is an officially accepted category in this journal or whether “Original Research” or “Short Communication” would be a more appropriate classification.
Dear Editorial Team,
Thank you for your email and for the opportunity to refine my review. I appreciate the importance of providing more detailed feedback to help the authors improve their manuscript. Below, I have answered your additional questions:
- What is the main question addressed by the research?
Study investigates the development and testing of an ICT tool for monitoring antibiotic use, animal diseases, and treatment outcomes in low-income countries. The goal is to enhance antimicrobial stewardship and surveillance capabilities in veterinary settings.
Do you consider the topic original or relevant in the field? Does it address a specific gap in the field?
Topic is relevant to the field of veterinary epidemiology and AMR. Given the growing concerns regarding AMR, particularly in low-resource settings, this study provides a timely and meaningful contribution. The application of ICT solutions for AMR surveillance remains an underexplored area, making this research particularly valuable.
What does it add to the subject area compared with other published material?
To the best of my knowledge, there have been no prior studies focusing on the development and implementation of an ICT tool for antimicrobial stewardship and disease monitoring in this specific region of Africa. While similar systems exist in other parts of the world, this study fills an important gap by addressing the unique challenges and conditions present in low-income countries within this part of the continent.
What specific improvements should the authors consider regarding the methodology? What further controls should be considered?
Methodology is well described and appropriate for the research aims.
Are the conclusions consistent with the evidence and arguments presented and do they address the main question posed?
Conclusions are supported by the data presented and align with the study’s objectives.
Are the references appropriate?
References are relevant and up to date.
Please include any additional comments on the tables and figures.
The figures and tables are well-structured and effectively summarize the study’s findings. If possible, the authors could consider including a visual representation of the ICT tool’s interface to help readers better understand its functionality.
Author Response
Thank you for your valid and useful comments. We have addressed each one in the responses below. We have also proof-read, edited and clarified the language as relevant.
Reviewer comment: The figures and tables are well-structured and effectively summarize the study’s findings. If possible, the authors could consider including a visual representation of the ICT tool’s interface to help readers better understand its functionality.
Response: Thank you for this suggestion, we have included a new figure (figure 2) that shows the user view of veterinary pharmacy staff and farmers, respectively.
Reviewer 2 Report
Comments and Suggestions for Authors
The manuscript is related to the monitoring of antimicrobials use and AMR in poultry based on a mobile application prototype from Kenya.
I have the following comments:
1. Why ADIS system utilize android application for collection of data from farmers whereas web-based application for veterinary pharmacies? Any specific reasons?
2. Data sets collected from farmer communities and veterinary pharmacies and subsequent data analysis are not shown.
3. Sample size is too small.
4. There is no information related to age-based stratification, educational levels of end users.
5. How can a correlation be established between sale of antimicrobials from veterinary pharmacies and their use for poultry by the farmers who were also the participants of same study?
6. There is little information on comparative analysis of ADIS system with already available android app/we tools developed in other regions of Africa and Europe.
7. What are the limitations of ADIS?
8. How the collected data are intended to be used for mitigating the menace of AMR in veterinary sector?
9. Data presentation is poor. Few tables, figures and app interfaces are essentially needed.
Comments on the Quality of English Language
May be improved
Author Response
Thank you for your valid and useful comments. We have addressed each one in the responses below. We have also proof-read, edited and clarified the language as relevant.
Reviewer comment: Why ADIS system utilize android application for collection of data from farmers whereas web-based application for veterinary pharmacies? Any specific reasons?
Response: It was assumed that the farmers would only have access to a mobile phone and as android phones are less expensive this was the first choice. As a web-based application would be easier for large amounts of data registration, this was the preferred option for veterinary pharmacies. This has now been clarified n the text (lines 288-291)
Reviewer comment: Data sets collected from farmer communities and veterinary pharmacies and subsequent data analysis are not shown.
Response: We understand this concern. However, these data will be presented in a separate publication, as we believe putting all this information into one publication would make the manuscript cumbersome and too heterogeneous. This other publication is almost ready for submission and has been announced to be submitted to another special issue of Antibiotics. It is referred to on line 430.
Reviewer comment: Sample size is too small.
Response: Yes, we agree, this is a valid point. If this would have been a study focusing on the collected data, the sample size would be far too small. As it was, we decided to pilot the system with a small number of participants to enable us to provide support and close follow-up throughout the pilot study period.
Reviewer comment: There is no information related to age-based stratification, educational levels of end users.
Response: Yes, thank you for this point. We did not include questions about the study participants during the piloting of the system. However, they were all from the previous baseline study (Mutua et al), referred to on line 418- 420, which was the target group. This is now further clarified in this section. We did not want to include questions on the users themselves at this stage as it might have deterred some participants and our main focus was on piloting the system.
Reviewer comment: How can a correlation be established between sale of antimicrobials from veterinary pharmacies and their use for poultry by the farmers who were also the participants of same study?
Response: There can be no direct correlation analysis as the pharmacies do not register their customers. In the future, if this or a similar system were to be used on national level, geographic information related to the input data (pharmacy location and farmer location) would allow for some such assessment on regional level. This has been added to the Discussion lines 235-242. Nevertheless, as stated on line 233-235, the system should be seen as a first step towards more detailed data collection and not as the final surveillance programme for collecting data on AMR/AMU (as this would also require registration of prescription data and sample collection from sick animals).
Reviewer comment: There is little information on comparative analysis of ADIS system with already available android app/we tools developed in other regions of Africa and Europe.
Response: Thank you, we would be grateful if you could provide examples of such systems. We have not been able to identify a similar tool that collects data from farmers and pharmacies in LMICs, but if directed to information about such a system we will be happy to add a comparison. We believe that comparisons with systems in HICs may be misleading, as the available resources, legislation and registers are so very different. We did try to find a ready-to use system when the project started but failed, which is the reason for developing the system in the country where it would be used. We realise that there may be similar projects underway, or being planned, and this is the reason for writing this paper, so that others may draw from our experiences.
Reviewer comment: What are the limitations of ADIS?
Response: As with all IT systems, there are many limitations, the main being that it relies on IT literacy, internet access and electricity for charging devices. Although it currently only includes some input data and basic information directed towards poultry farmers, it can be developed to a broader target group. This was highlighted in the Discussion on lines 191-194 but more has been added on lines 229-230 and 250-253
Reviewer comment: How the collected data are intended to be used for mitigating the menace of AMR in veterinary sector?
Response: In the long-term, the data are intended to support national authorities in providing a picture on antimicrobial use (combining data on sales and drugs administered on farm level), both as regards the amounts used, the reason for use and the outcome of this use. Such data on national and regional level will help design policies (legislation and control as well as information campaigns). This has been clarified on lines 235-242.
Reviewer comment: Data presentation is poor. Few tables, figures and app interfaces are essentially needed.
Response: As explained above, we have not included the collected data and as the sample size is small and data are mainly qualitative, we chose to present results in narrative format. We have added a figure (figure 2) demonstrating the user view of veterinary pharmacy staff and farmers, respectively, thank you for this suggestion.
Reviewer 3 Report
Comments and Suggestions for Authors
Reviewer Comments
- Please mention the existing method of documenting the drug sales and if and how the proposed ICT tool could be integrated with it.
- Is a better cataloguing system for their drug sales the only benefit for the drug stores? Will they eventually be weaned off the system? Are they getting burdened with this additional work? Please include other possible benefits to the pharma stores to document the drug sales details until the time when it becomes mandated by the state.
- Likewise, please clearly mention the advantages the farmers would derive from the tool. Is it only the knowledge sharing that the tool talks of? Is the information something the farmer has to read or is it voice enabled AI? This is important as the farmer might not have the time to read the information or might be illiterate.
- Is the tool enabled only in English or any local language that the farmer is comfortable with? If it is only in English is there a translate option?
- How was the data collected and compiled? Based on the Animal/ Disease /Drug?
- Please include the % success when it comes to connecting with a veterinarian listed in the tool?
- Please clearly mention how the issue of slow internet connectivity in remote places would not offset the tool usage.
- Please also include the backend data security measures.
- Please expand FGDs initially as the expansion comes in the materials and methods section which comes later.
- Results - Lines 165-166: After demonstrating the ITC system, most reactions were positive. / After demonstrating the ICT system, most reactions were positive.
Author Response
Thank you for your valid and useful comments. We have addressed each one in the responses below. We have also proof-read, edited and clarified the language as relevant.
Reviewer comment: Please mention the existing method of documenting the drug sales and if and how the proposed ICT tool could be integrated with it.
Response: Thank you for this suggestion. Currently, pharmacies document drug sales by hand-written notes in a ledger. If data were to be primarily entered in ADIS or a similar system this would allow for own follow-up on stocks and financial input as well as for authorities’ control. We have added this information on line 77-78
Reviewer comment: Is a better cataloguing system for their drug sales the only benefit for the drug stores? Will they eventually be weaned off the system? Are they getting burdened with this additional work? Please include other possible benefits to the pharma stores to document the drug sales details until the time when it becomes mandated by the state.
Response: Again, thank you for highlighting this, we have included a comment on this in the Discussion, on line 196-200.
Reviewer comment: Likewise, please clearly mention the advantages the farmers would derive from the tool. Is it only the knowledge sharing that the tool talks of? Is the information something the farmer has to read or is it voice enabled AI? This is important as the farmer might not have the time to read the information or might be illiterate.
Response: Thank you for this valid point, it is challenging to incentivise farmers. Our hope was that the farmers would see the benefits of the information and veterinary contacts so that they would regularly use the system. Our recruited farmers were all literate and fluent in English and the messages in the information were written as clearly and simply as possible, in brief points and organised so that specific information would be easy to locate when seeking it. In the future, voice enabled AI could be added, as well as some requirement for regular data entry to be able to access the system. We have added this to the Discussion (line 248-252) and the M&M (lines 355-357).
Reviewer comment: Is the tool enabled only in English or any local language that the farmer is comfortable with? If it is only in English is there a translate option?
Response: Currently the system is only available in English, as this is spoken by all farmers in the target population (i.e. the participants in the baseline study, see Mutua et al, 2023 as referred to on line 418). If used for other farmer populations, the texts would need to be translated. This is clarified in the addition to the Discussion (line 250).
Reviewer comment: How was the data collected and compiled? Based on the Animal/ Disease /Drug?
Response: Pharmacies enter data based on drug and drug package, whereas farmers enter data based on individual animal treatments. This has been clarified in the M&M line 286-288
Reviewer comment: Please include the % success when it comes to connecting with a veterinarian listed in the tool?
Response: This is a valid point and we should have collected such information. Sadly, we did not so it is not available. We have now stated this on line 260-261.
Reviewer comment: Please clearly mention how the issue of slow internet connectivity in remote places would not offset the tool usage.
Response: We have added a sentence on this on line 231-232.
Reviewer comment: Please also include the backend data security measures.
Response: This has been added to the M&M section, lines 342-350.
Reviewer comment: Please expand FGDs initially as the expansion comes in the materials and methods section which comes later.
Response: We have clarified the acronym and added some additional information to the Results (line 113-114).
Reviewer comment: Results - Lines 165-166: After demonstrating the ITC system, most reactions were positive. / After demonstrating the ICT system, most reactions were positive.
Response: Thank you, this has been corrected.
Round 2
Reviewer 2 Report
Comments and Suggestions for Authors
Author's replies to the comments are relevant and convincing. The intent was to pilot test the App and the compilation of subsequent larger data sets is underway. The initiative is a commendable step towards root level monitoring of antimicrobials uses in LMICs. This could prove to be a cutting edge ICT-enabled intervention to report & curtail the rapid emergence of AMR and ARBs in poultry and veterinary sector in the context of One Health.
I am of opinion that the study is significantly important and deserve publication.